# Influence of Gender, Age and Field of Study on Hand Hygiene in Young Adults: A Cross-Sectional Study in the COVID-19 Pandemic Context

**DOI:** 10.3390/ijerph182413016

**Published:** 2021-12-09

**Authors:** Maria Barcenilla-Guitard, Anna Espart

**Affiliations:** 1Nursing Area, Hospital Universitari Arnau de Vilanova, 25198 Lleida, Spain; mariabg98@gmail.com; 2Department of Nursing and Physiotherapy, Serra Húnter Lecturer, University of Lleida, 25198 Lleida, Spain; 3Faculty of Nursing and Physiotherapy, University of Lleida, 25198 Lleida, Spain; 4Development of Healthy Organizations and Territories (DOTS), 25001 Lleida, Spain; 5Research Group of Health Care (GRECS), Lleida Institute for Biomedical Research, Dr. Pifarré Foundation, IRBLleida, 25198 Lleida, Spain

**Keywords:** hand hygiene, hand washing, hand hygiene knowledge, hand hygiene attitudes, hand hygiene practice, transmission of infectious disease, young adults, health literacy, COVID-19

## Abstract

The effectiveness of hand hygiene (HH) on reducing the transmission of contagious diseases is widely known, although its use has been commonly associated with the area of healthcare. During the COVID-19 pandemic, HH was one of the main measures established to contain the transmission of this virus. The identification of the main barriers and facilitators of HH in young adults (aged 18–29 years old) will contribute to the better planning of HH training and its posterior success. A total of 716 young adults participated in the study by completing the ad hoc online questionnaire (*#YesWeHand*), which analyzed, among other aspects, the age range, gender and field of study that they belonged to. From the total participants, 81.3% indicated knowing how to perform HH correctly, while 49.4% affirmed having received training. The main reason for performing HH was concern for their own safety and that of others (75.8%), while forgetfulness (36.5%) was the main reason for not performing HH. In the group of young adults, being female, aged between 22 and 25 years old, and having studied in the area of Health Sciences, had a positive influence on correct HH. It is deemed necessary to maintain HH beyond the primary education stages, and to adapt it to different fields of education, ages, and genders, in order to maximize its success. Given the overrepresentation of participants from the healthcare field, it would be desirable to conduct more studies to ensure a better representation of the different educational levels and fields of study of the participants, in order to identify, in a more reliable way, the variables that influence HH.

## 1. Introduction

The effectiveness of correct hand hygiene (HH) for decreasing infectious diseases has been widely known since Ignaz Semmelweis demonstrated the importance of adequate hand washing for the reduction of puerperal fever mortality in the 19th century [1,2]. In spite of this, since its discovery, this practice has been mainly applied in the area of health care, as an essential method for the reduction of infectious diseases, as well as in primary schooling, to teach children about correct hygienic health habits [3,4]. 

### 1.1. Hand Washing and COVID-19

Due to the SARS-CoV-2 pandemic, declared by the World Health Organization (WHO) in March 2020, the importance of correct hygiene has become one of the fundamental health measures, along with the use of masks and social distancing, established by a great number of nations affected by this virus, to contain COVID-19 cases in the population [5,6,7]. In spite of this, although the use of masks and the maintenance of social distancing can be objectively controlled by those in charge (volunteers, police, security agents, etc.), HH, due to the characteristics of its process (that is, the process of doing so, technique used, material necessary, etc.), may not have been conducted adequately enough to be effective on the reduction of virus particle transfer from fomite surfaces to the hands and from these to the nose membranes, mouth, and eyes [8,9]. However, a correct HH can not only help us in the prevention of SARS-Co-V-2 disease, but is also essential in avoiding other diseases that may require antibiotic treatment; accordingly, the usual practice of HH also contributes to a lower number of cases of antibiotic resistance [10].

During the pandemic, numerous actors (health professionals, scientists, health institutions, teachers, and communication media, among others) contributed to presenting and promoting the importance of HH techniques outside of the field of health care, by utilizing hand washing, or HH with a water-alcohol solution, which has been normalized as an effective method to a correct HH since the 2009 H1N1 influenza pandemic [11]. Among these two techniques, hygiene with a water-alcohol solution was the most utilized worldwide, thanks to its easy application, at the access points at every type of establishment, and the lesser requirement of material for performing this technique [12,13]. 

Despite the availability of information to the general population about how to correctly perform HH, not only in relation to the COVID-19 pandemic but also before this, access to information *per se* is not enough, as it this information needs to be to be oriented and adapted to different social and generational groups to ensure its success and the implementation of correct HH [14]. 

### 1.2. Young Adults and COVID-19

In Spain, as in most countries affected by COVID-19, different actions and measures were implemented from the start of the pandemic in order to avoid the transmission of the virus. Starting in May 2021, the third state of alarm declared by the Spanish government came to an end [15]. This implied, among other aspects, the return of the free circulation and movement of the population at any time and place in Spain. Aside from this measure, in June 2021, the non-mandatory nature of the use of masks in public outdoor spaces was declared. These two actions resulted in the abandonment of two of the three individual actions demanded from the population (masks and the maintenance of social distance between people, in the public outdoor spaces). These changes, along with the greater presence of the Delta variant, resulted in the appearance of the fifth wave of infections, baptized as the *young wave*, due to the great repercussion among adolescents and young adults in terms of community transmission, as many people in this demographic had not been vaccinated at that time, as they were not part of the vaccination group, according to the protocols established [16].

The population described as young adults, those aged between 18 and 29 years old, despite their having reached full biological, physical, and psychological faculties, and having a great degree of independence in making their own decisions, are considered a heterogeneous group. Within this group, we find young people who did not continue their formal education beyond mandatory education, those who continued with non-mandatory education (technical school or university), and those who were already working. Thus, we find a large number of differences in terms of their knowledge, attitudes, and practices related to different aspects associated with health and its risks, due to their age, gender, and their field of study, given their education or training up to this point [14,17].

In a context of greater awareness, such as the COVID-19 pandemic, the identification of differences in the performance of a basic process in health prevention, such as the correct performance of HH in a highly heterogeneous population, such as young adults, could be useful when planning actions associated with health literacy and health education programs that are more efficient for this and other population groups. Thus, starting with three variables, such as age, gender, and field of education, we seek to identify if these have an influence on the knowledge, practice, and attitudes of young adults in relation to HH habits. 

### 1.3. Aim of the Study

The objective of the study was to identify the knowledge, practices, and attitudes of young adults, at the international level, and their relationship with age, gender, and field of education, which could have an influence as barriers or facilitators for correct HH, in the pandemic context of disease due to coronavirus (COVID-19).

## 2. Materials and Methods

### 2.1. Study Design and Participants

For this study, an observational, descriptive and cross-sectional design was utilized through the use of an online questionnaire in order to identify the knowledge, practices, and attitudes of the study participants, related to HH. Based on snowball sampling, the main author contacted people close to her, using WhatsApp® (Meta Platforms Inc., Menlo Park, CA, US) and Telegram® (Telegram Messenger Inc., London, USA) contacts to participate in the study, and the participants themselves were asked to disseminated the questionnaire among their acquaintances. Likewise, content creators from different fields (related and unrelated to the health field) were contacted to publicize the study and recruit more participants.

The participants came from an international group of young adults (18–29 years old) who agreed to participate in the study by self-completing the online questionnaire. As it was impossible to have access to a representative sample of young adults worldwide, a convenience sampling method was utilized, which allowed us to gather information about habits, opinions, and points of view.

The participants accessed the online questionnaire through the different digital means through which the study was disseminated, and through which they were not only asked to participate, but to also disseminate the questionnaire among their acquaintances in order to reach a larger population of potential participants.

### 2.2. Design, Distribution and Access to Questionnaire

*The #YesWeHand* self-completed questionnaire was designed starting with the relevant information from other studies related to the key aspects of correct and effective HH [17,18,19]. The 31 questions included in the questionnaire were mostly simple choice (24 questions), multiple choice (5 questions), Likert-type (1 question), and open-ended (1 question). The different questions belonged to the four areas to be analyzed: first, sociodemographic aspects (including questions related to age, gender, and field of education); second, questions centered on knowledge about HH; third, questions oriented to discover HH practices; and fourth, specific questions on the main attitudes related to the importance of HH. The questionnaire was previously tested to detect possible dysfunctions or comprehension errors before it was deemed as apt for data collection. The mean estimated time for self-completing the questionnaire was five minutes.

For recording and managing the responses, the designed questionnaire was transferred to the questionnaire administration program Google Forms^®^ (Google LLC, Mountain View, CA, US). The questionnaire was created in English, Spanish, French, Italian, and Portuguese in order to obtain a greater number of responses and to avoid the limitations due to language as an exclusion criterion in the study. On the first screen of the online questionnaire, the participant could select the language in which to participate in the study, and from this, he or she could access the questionnaire in their desired language.

To provide the questionnaire to the potential participants, the two most popular instant messaging distribution channels worldwide were utilized: WhatsApp^®^ and Telegram^®^, as well as the social networks Instagram^®^ (Meta Platforms Inc., Menlo Park, CA, US) and Facebook^®^ (Meta Platforms Inc., Menlo Park, CA, US). To increase the reach of the study, different Instagram^®^ content creators, who had a large number of followers, were invited to disseminate the study and the questionnaire to maximize the number of participants.

### 2.3. Data Analysis

Descriptive statistics were utilized to show the sociodemographic and main characteristics related to the knowledge, practices, and attitudes of the participants. For the analysis of the association between the variables of interest (age, gender, and field of study) and the rest of the variables analyzed, the Chi-square test or Fisher’s exact test were utilized, and to find the effect size of the association, Cramer’s V was calculated (whose effect size is identified as: 0.00–0.09 negligible, 0.10–0.19 weak, 0.20–0.39 moderate, 0.40–0.59 relatively strong, 0.60–0.79 strong and 0.80–1.0 very strong). A level of significance of *p* < 0.05 was utilized for all of these tests. To process the data, the IBM^®^ SPSS^®^ (IBM Corp., Armonk, NY, US) version 26 statistics program for Macintosh was utilized.

### 2.4. Ethical Aspects

Before starting the questionnaire, on the first screen, the participants were informed about the study, its aim, and the confidential and anonymous nature of all their answers, in agreement with the General Data Protection Regulation from the European Union (from 25 May 2018). The participants were also asked for their prior consent before starting the questionnaire to be able to record their responses.

## 3. Results

The responses obtained with the questionnaire were collected between January and March, 2021. The responses from a total of 716 participants were collected, all of which were valid for data analysis. Gender was identified as female, male or non-binary. Age was grouped in three bands: young adults who continued with non-mandatory education (18–21 years old), young adults who have completed their non-mandatory education and are starting in the labor market (22–25 years old), and young adults who have been in the labor market for a few years (26–29 years old). Field of study was identified as Arts and Humanities, Engineering and Architecture, Social Sciences and Law, Health Sciences, and Exact, Natural and Life Sciences.

### 3.1. Sociodemographic Characteristics of the Participants

In the results analyzed, the percentage participation of women was notable, being four times higher (80.6%) than that of men. The 22–25 age group was identified as the most participative (41.1%). In addition, the percentage of participants who identified themselves as students at the time the data was recorded was higher (40.1%) than the rest of the groups, with the field of Health Sciences being the most identified as their field of education (59.4%). The majority of responses came from Spain.

When asked about the COVID-19 context, a great majority indicated not having been infected (81.3%), more than half also indicated that they did not have family members who had contracted the disease (57.5%), and half of them (50.4%) indicated that they did not currently live with people at risk of becoming infected with the virus (Table 1).

### 3.2. Knowledge about Hand Hygiene

As for the results related to HH knowledge possessed by the participants, a large majority indicated knowing how to correctly clean their hands (81.3%), knowing the basic steps (94.3%), and having enough knowledge about it (70.0%), in spite of only half of them indicating having received specific training on this (49.4%). Among the questions asked that were intended to evaluate whether the knowledge they possessed was adequate, a third (33.1%) of the participants ensured that HH eliminated between 90–95% of the microorganisms, despite 98.6% of them ensuring that HH was effective in the prevention of infectious diseases. Likewise, only a small percentage (6.1%) identified the water-alcohol solution as the most effective method for HH.

It should be highlighted that a statistically significant association between the field of study of the participants and the answers recorded was clearly evident (*p* < 0.001 and <0.031). In most cases, the relationship established between these cases was moderate between variables (effect size > 0.20 and <0.41), although it should be highlighted that there was a relatively strong association between the field of study and having received training on HH (*p* = 0.000; effect size = 0.59). Age and gender were not highlighted for having a statistically significant relationship with the variables analyzed, aside from a weak association between age and knowledge of the number of existing steps for correct HH (*p* = 0.009; effect size = 0.13), having enough knowledge (*p* = 0.000; effect size = 0.17) or knowing the percentage of microorganisms eliminated with hand washing (*p* = 0.006; effect size = 0.18) (Table 2).

When asked about the type of microorganisms that the water-alcohol solution is able to eliminate, the majority of participants identified bacteria (75.4%) as the main microorganism that could be eliminated, followed by viruses (67.4%), and in a smaller percentage, fungi (27.5%). Likewise, 12.0% of the participants indicated that they were not clear on what microorganisms could be eliminated with the water-alcohol solution. In every case, a statistically significant relationship was established, with an effect size between weak and moderate (effect size > 0.18 and <0.25), between the answer provided and the field of study of the participants (Table 3).

Lastly, with respect with the main sources of information utilized to obtain knowledge about correct HH, they highlighted health professionals (80.3%), and scientific articles (52.1%) as the main sources of information. However, 31.7% of the participants also indicated social networks, television (25.3%), or webpages (24.2%). Once again, statistically significant relationships (*p* = 0.000) were established between the field of study and the different sources consulted, with a relationship of moderate effect established (effect size > 0.19 and <0.34), except for the relationship established between the field of study and the use of health professionals as a source of information, for which a weak relationship was found (*p* = 0.002; effect size = 0.16) (Table 4). 

### 3.3. Attitudes about Hand Hygiene

As for the attitudes of the participants with respect to HH, it was observed that the number of times they indicated performing HH was weakly associated to the age group they belonged to (*p* = 0.001; effect size = 0.17). With respect to the main reason which motivated them to perform HH, *For my safety and that of others*, statistically significant relationships were observed with the age group they belonged to (*p* = 0.02; effect size = 0.19), with their gender (*p* = 0.022; effect size = 0.16), and their field of study (*p* < 0.001; effect size = 0.28). Lastly, with respect to the reasons for not performing HH when they had to, the most common were *I do not remember* (36.5%) and *I do not have the necessary material* (29.7%). After analyzing the relationship established between the different reasons that impeded them from performing HH, with the variables of interest (age, gender, and field of study), the field of study was identified as the variable which influenced the greatest number of reasons, followed by age and gender. In practically every case, the effect size of the existing relationship between the different variables was weak, except for the relationship between the reasons for performing HH, and the field of study of the participants (Table 5).

### 3.4. Practices about Hand Hygiene

As for the practices applied for HH, 76.1% of the study participants ensured doing these correctly, although when asked about the amount of time utilized for hand washing with water and soap, only 36.5% answered that they took between 40 and 60 s, and only 22.9% indicated between 20 and 30 s when the hygiene was performed with a water-alcohol solution (both of which are recommended for correct HH, according to the WHO). Again, the field of study was identified as having a statistically significant relationship on the different practices applied, with the effect size varying between weak and moderate (effect size > 0.12 and <0.39) (Table 6).

When the participants were asked about the different steps taken when washing their hands, while the first three steps were applied by a large percentage of the participants, step 4, rubbing of the dorsal part of the fingers, was performed by less than half of the participants (48.2%), and step 5, rubbing of the thumbs, and step 6, rubbing of the palms, by slightly more than half of participants (55.0% and 58.4%, respectively). Despite the observation that, in some of the steps, there was a statistically significant relationship between the steps taken and the age group or gender of the participants, it was again the field of study variable which was more frequently statistically associated with the performance of these steps, with the predominance of a moderate effect size (effect size > 0.15 and <0.33) (Table 7).

### 3.5. Facilitators and Barriers for Proper Hand Hygiene

Among the different aspects included in the *#YesWeHand* questionnaire, three of these were about knowledge, one was about attitudes, and two were about the practice. This allowed us to determine these as the main facilitators and barriers for and against the correct hygiene of hands. As facilitators, we identified being part of the 22 to 25 years old group, being female, and being in the Health Sciences field of study. On the contrary, as barriers, we identified any of the other two age groups (18–21 or 26–29 years old), being a male, and currently studying or having studied in the Arts and Humanities field of study, and to a lesser extent, studying or having studied in the field of Engineering and Architecture (Table 8 and Table 9).

## 4. Discussion

In the present study, the main barriers and facilitators for the correct HH of a group of young adults were identified. Starting with the set of responses obtained during the COVID-19 pandemic, in which a greater awareness existed towards correct HH, significant data were observed in relation to gender, age, and field of study. Despite some studies identifying gender, age, and level of education as having an influence on better knowledge, attitudes and practices with respect to HH in the general population, adolescents or school-aged children, as far as we know, this is the first study which exclusively focuses on young adults and also includes the influence of their field of study [19,20,21,22,23].

Although the study was designed to be an international study, most of the responses came from young adult females from Spain. This is perhaps because these participants shared the questionnaire with their closest acquaintances with similar sociodemographic characteristics, despite the questionnaire being disseminated in different languages and on various popular platforms. There are 7% of participants who have not indicated the field of study; this is very possibly due to the fact that there is no specific training area in compulsory education. Despite this, they would only represent 50 participants of the 200 who have indicated that their highest level of education is compulsory education. That is why the rest of the participants (150) would identify themselves within one of the study areas, either because they have completed some specific training at some point or because their work area that they identify is included within one of these proposed areas.

Likewise, most than half of the participants indicated that they are studying or have studied in the field of Health Sciences, which could explain why most of the individuals who completed the questionnaire also came from the same field; it is also possible that this was because individuals who had studied in the Health Sciences field were more inclined to answer the questionnaire. 

The self-perception of having enough knowledge (70.0%), of correctly performing the task (76.1%), and of knowing all the different steps of HH (94.3%) differed from the considerably lower percentages of participants who indicated the percentage of microorganisms eliminated with HH (33.1%), the most effective method (6.1%), and the correct timing of hand washing with soap and water (36.5%) or disinfection with a water-alcohol solution (22.9%). All of these findings corroborate the results by Kitsanapun et al. regarding the importance of persevering with the training and the correct performance of HH, beyond one-time training programs [24]. This demonstrates that, when individuals believe that their perception is correct, they stop updating or improving their knowledge and practices.

With respect to the sources utilized by young adults, in our study we identified health professionals, scientific articles, and social networks as the main sources of information related to correct HH. These data partially coincide with other studies, in which social networks and internet searches are identified as the main sources used by university students to find out about health-related topics [25,26]. This differs from other studies, in which family members and traditional communication media were the most utilized to increase one’s health literacy [27]. These results, which differed from other studies, could be explained by the social desirability bias, and by the fact that most of the participants came from the field of Health Sciences.

Despite the women indicating that they possessed less knowledge about HH, they stated that they performed it more frequently than necessary, stating that they did so for their safety and that of others (data not shown). These data are in agreement with the study by Chen et al., where it was observed that women performed HH 1.12 more times than men [23]. Likewise, other authors have also described the female gender as having more awareness about personal safety and that of others in the performance of health prevention actions [28,29,30,31]. On the contrary, a significant percentage indicated that the two main reasons why they do not carry out HH is because *I do not remember* and because *I do not have the necessary material*. In the context of the COVID-19 pandemic, in which the presence of alcoholic solution is common in any place, it is possible that the participants who have indicated that they lack the necessary material for a good HH are those who consider that water and soap are the necessary materials for the correct HH and therefore, not being able to carry out hygiene with soap and water in all places, they consider that they lack the necessary material.

As for the age groups analyzed in this study, less knowledge was observed in the 18 to 21 and 26 to 29 age groups, which indicates that knowledge about HH increased after the age of 25, although from this point on, this knowledge was gradually forgotten. The same occurs with attitudes and practices; even if these improved until the 22–25 age group, a deterioration was detected as the individual ages. This re-enforces the observations from other authors, who described a loss of knowledge, practices, and attitudes acquired related to HH as time passed, without the re-enforcement of education and training on the subject [32,33].

As for the field of education, and as predicted, the participants educated in the field of Health Sciences showed better knowledge, practices, and attitudes as compared to the rest of the fields of study, with those in the fields of Engineering and Architecture the ones who possessed less knowledge, and those in the fields of Arts and Humanities, the ones who showed, to a greater degree, non-adequate attitudes and practices. All of this only corroborates the need for cross-disciplinary training on health literacy in the different fields of study. Education could result in the improvement of prevention in different aspects of health at the level of society [34,35].

### 4.1. Implications

The main implications of the study are to consider differences in age, gender, and field of study. Given that age was identified as a key factor, and a lower level in the indicators of knowledge, practices and attitudes was identified in older age ranges (i.e., 26–29 years), this factor should be taken into account when designing strategies to improve HH indicators. Establishing health literacy actions in the area of HH is necessary, so that these can be applied continuously, not only as one-time training, to guarantee more successful results and, therefore, obtain better results in the reduction of cross-contamination. 

Similarly, given the overrepresentation of participants from the health field, it should be taken into account that the design of strategies to improve HH will have to be specific to this sector, which already receives specific training during their education. In the case of participants who are trained in fields other than health, a deeper knowledge of knowledge, practices and attitudes will be required, so that the strategies designed can be adapted to their field of training.

### 4.2. Limitations

In this study, there was an over-representation of females, from the field of Health Sciences, and of Spanish nationality. This could be the result of the use of instant messaging apps and social networks for the dissemination of the study and questionnaire. The over-representation of some sectors of the population makes extrapolation of the results to the entire group of young adults difficult.

On the other hand, the anonymous nature of the questionnaire does not avoid the bias of social desirability in which the participants demonstrate desirable behaviors, which could be far different from those that they exhibit in their life.

Given that the study utilized a self-completed questionnaire, it is possible that on some questions regarding the practice of HH, there could be differences between participants, even when applying the same actions, which could result in the appearance of subject biases. In future studies, it would be interesting, for example, to include a question about HH knowledge that includes a false answer to determine whether the participants select it as correct consciously or by random, and thus better discriminate the possible knowledge they possess.

## 5. Conclusions

Our study demonstrates that young adults perceive themselves as having enough knowledge about HH and how to perform it correctly, despite the fact that only half of them indicated having received some type of specific training for this. The individuals educated in the field of Health Sciences have a greater degree of knowledge with respect to people educated in other fields, although they also have deficiencies in their knowledge, practices, and attitudes. These deficiencies are greater in individuals educated in other fields of knowledge.

Being a female implies a greater degree of knowledge, attitudes, and better practices related to HH; while the male gender is less involved in any of the three areas, which could result in the decreased protection of this gender against cross-contamination which exists with direct hand contact. Given that there are important differences between the both genders, as well as depending on the field of study, it would be necessary to explore the best strategies to improve the indicators related to correct HH (i.e., knowledge, practices and attitudes) in men, and in those not trained in the field of Health Sciences.

Age was found to be a key factor; thus, as one becomes older, there is an increase in knowledge and an improvement in the practices and attitudes until a certain age limit. From this point on, age works against us, and has a negative influence on the maintenance of knowledge, the practices, and attitudes. This aspect could be more worrying in individuals educated in the field of Health Sciences, as this could drive them away from the correct health care practices, which are necessary for protecting the health of patients and one’s own health.

It is necessary to include and improve education on HH through the early education period and in different fields of higher education beyond the Health Sciences.

Much information can be found about the need to perform HH, but not about how to do so properly. It is assumed that as this is a common, everyday practice, everyone does it correctly, so that it is important to re-enforce the knowledge, practices, and the best attitudes throughout the years, and it is also necessary to involve the male population in the process of HH, if we want to maximize its positive effects on society.

## Figures and Tables

**Table 1 ijerph-18-13016-t001:** Sociodemographic characteristics of participants.

Variables	N = 716
Gender	n (%)
Female	577 (80.6)
Male	134 (18.7)
Non-binary	5 (0.7)
Age, mean (± SD)	23.5 (3.31)
Age (group)	n (%)
18–21	209 (29.2)
22–25	294 (41.1)
26–29	213 (29.7)
Continent of residence	n (%)
Europe (excluding Spain)	22 (3.1)
Spain ^a^	662 (92.4)
Americas (all countries)	32 (4.5)
Others	0
Educational level achieved	n (%)
Compulsory education	200 (27.9)
Professional training	152 (21.2)
University degree or above	364 (50.8)
Field of study	n (%)
Arts & Humanities	35 (4.9)
Engineering & Architecture	51 (7.1)
Social Sciences & Law	105 (14.7)
Health Sciences	425 (59.4)
Exact, Natural & Life Science	50 (7.0)
Studying or working?	n (%)
Studying	287 (40.1)
Working	211 (29.5)
Studying & working	204 (28.5)
Not studying not working	14 (1.9)
Have you had COVID-19?	n (%)
Yes	105 (14.7)
No	582 (81.3)
Not sure	29 (4.1)
Did someone in your family have COVID-19?	n (%)
Yes	287 (40.1)
No	412 (57.5)
Not sure	17 (2.4)
Living with people at risk for COVID-19? ^b^	n (%)
Yes	286 (39.9)
No	361 (50.4)
Not sure	69 (9.7)

a: Spain has been excluded from the Europe count due to the large number of responses from this country; b: Including elderly, people with comorbidities (diabetes, hypertension, cardiovascular diseases, chronic respiratory diseases), oncologic patients, pregnant and obese people.

**Table 2 ijerph-18-13016-t002:** Knowledge self-reported by the participants in relation to hand hygiene.

Self-Reported Knowledge				
	*n* (%)	Age *p*-Value (Effect Size)	Gender *p*-Value (Effect Size)	Field of Study *p*-Value (Effect Size)
I know how to perform proper HW				
Yes	582 (81.3)	--	--*	<0.001 (0.40-moderate)
No	91 (12.7)			
Not sure	43 (6.0)			
I know the steps of HW				
Yes	675 (94.3)	0.009 (0.13-weak)	--*	<0.001 (0.34-moderate)
No	27 (3.8)			
Not sure	14 (2.0)			
I have enough knowledge about proper HW				
Yes	501 (70.0)	0.000 (0.17-weak)	--	0.000 * (0.37-moderate)
No	178 (24.9)			
Not sure	37 (5.1)			
I have learned about HW in workshops				
Yes	354 (49.4)	--	--	0.000 * (0.59-rel. strong)
No	353 (49.3)			
Not sure	9 (1.3)			
Does the use of gloves replace HW?				
Yes	11 (1.5)	--	--	0.010 (0.18-weak)
No	698 (97.5)			
Not sure	7 (1.0)			
What percentage of microorganisms is eliminated with proper HW?				
Less than 25%	2 (0.3)	0.006 * (0.18-weak)	--*	0.030 * (0.24-moderate)
25–50%	10 (1.4)			
50–80%	142 (19.8)			
80–90%	176 (24.6)			
90–95% ^T^	237 (33.1)			
More than 95%	116 (16.2)			
Not sure	33 (4.6)			
Is HW effective in preventing illness?				
Yes	706 (98.6)	--	--	0.016 (0.21-moderate)
No	5 (0.7)			
Not sure	5 (0.7)			
The most effective HW method I think is…				
water + soap	524 (73.2)	--*	--*	0.028 (0.22-moderate)
water-alcohol solution ^T^	44 (6.1)			
both are equally effective	141 (19.7)			
not sure	7 (1.0)			

HW: hand washing. T: true answer. “--“: not statistically significant. * Fisher’s exact test was applied.

**Table 3 ijerph-18-13016-t003:** Types of microorganisms eliminated with a water-alcohol solution, according to the participants (multiple choice answer).

	*n* (%)	Age *p*-Value (Effect Size)	Gender *p*-Value (Effect Size)	Field of Study *p*-Value (Effect Size)
Viruses ^T^	484 (67.6)	--	--	0.000 (0.24-moderate)
Bacteria ^T^	540 (75.4)	--	--	0.000 (0.19-weak)
Fungi ^T^	197 (27.5)	--	--	0.000 (0.20-weak)
None of the above	8 (1.1)	--	--	--
Not sure	86 (12.0)	--	--	0.000 (0.21-moderate)

T: true answer. “--“: not statistically significant.

**Table 4 ijerph-18-13016-t004:** Main sources of information about proper hand hygiene.

	*n* (%)	Age *p*-Value (Effect Size)	Gender *p*-Value (Effect Size)	Field of Study *p*-Value (Effect Size)
Family	98 (13.7)	--	--	--
Friends	51 (7.1)	0.040 (0.09-negl)	--	0.000 (0.23-moderate)
Health professionals	575 (80.3)	--	--	0.002 (0.16-weak)
Scientific papers	373 (52.1)	--	0.007 (0.11-weak)	0.000 (0.32-moderate)
Web pages	173 (24.2)	--	--	0.000 (0.30-moderate)
Newspapers	25 (3.5)	0.008 (0.11-weak)	--	0.000 (0.23-moderate)
Television	181 (25.3)	0.006 (0.11-weak)	--	0.000 (0.33-moderate)
Social networks	227 (31.7)	0.009 (0.11-weak)	--	0.000 (0.22-moderate)
Not sure	16 (2.2)	--	0.014 (0.12-weak)	0.000 (0.20-moderate)

negl: negligible. “--“: not statistically significant.

**Table 5 ijerph-18-13016-t005:** Attitudes self-reported by the participants in relation to hand hygiene.

**Self-Reported Attitude**				
***n* (%)**		**Age** ***p*-Value** **(Effect Size)**	**Gender** ***p*-Value** **(Effect Size)**	**Field of Study** ***p*-Value** **(Effect Size)**
I wash my hands…				
more than necessary	173 (24.2)	0.001 (0.17-weak)	--*	--
as often as necessary	300 (41.9)			
less than necessary	242 (33.8)			
Why do I wash my hands? (main reason)				
For my safety For the safety of others For my safety and that of others Out of habit Out of obligation None of the above	37 (5.2) 19 (2.7)	0.002 * (0.19-weak)	0.022 * (0.16-weak)	<0.001 (0.28-moderate)
	543 (75.8)			
	114 (15.9)			
	2 (0.3) 1 (0.1)			
Why do I not wash my hands? (multiple choice answer)				
My hands get damaged	66 (9.2)	0.039 (0.09-negl.)	--*	0.007 * (0.15-weak)
I usually keep them clean	94 (13.1)	--	--*	--*
Lack of time	121 (16.9)	--	--*	0.039 * (0.12-weak)
I do not have the necessary material	213 (29.7)	0.016 (0.10-weak)	--*	--*
I do not remember	261 (36.5)	<0.001 (0.13-weak)	0.005 (0.12-weak)	0.003 * (0.16-weak)
Lack of knowledge	7 (1.0)	--*	--*	0.011 * (0.19-weak)
It is not necessary	39 (5.4)	--	--*	--*
Others	66 (9.2)	--	--*	--*

negl: negligible. “--”: not statistically significant. * Fisher’s exact test was applied.

**Table 6 ijerph-18-13016-t006:** Practices self-reported by the participants in relation to hand hygiene.

Self-Reported Practice				
	n (%)	Age *p*-Value (Effect Size)	Gender *p*-Value (Effect Size)	Field of Study *p*-Value (Effect Size)
I perform proper HW				
Yes	545 (76.1)	--	--*	<0.001 (0.28-moderate)
No	73 (10.2)			
Not sure	98 (13.7)			
How much time do you spend on the whole HW process with water + soap?				
Less than 20 s	95 (13.3)	--*	--*	<0.001 (0.13-weak)
20–40 s	253 (35.3)			
40–60 s ^R^	261 (36.5)			
More than 60 s	104 (14.5)			
Not sure	3 (0.4)			
How much time do you spend on the whole HW process with a water-alcohol solution?				
Less than 10 s	152 (21.2)	--*	--*	0.000 (0.38-moderate)
10–20 s	252 (35.2)			
20–30 s ^R^	164 (22.9)			
30–40 s	90 (12.6)			
40–50 s	32 (4.5)			
50–60 s	10 (1.4)			
More than 60 s	9 (1.3)			
I do not know	7 (1)			
How long do you lather during hand washing?				
5–15 s	184 (25.8)	--	--*	<0.001 (0.30-moderate)
15–30 s	276 (38.6)			
30–45 s	171 (23.9)			
45–60 s	85 (11.7)			
Do you dry the hands after HW?				
Always	522 (72.9)	--*	--*	--*
Almost always	170 (23.7)			
Almost never	23 (3.2)			
Never	1 (0.1)			

HW: hand washing. R: recommended time. “--”: not statistically significant. *: Fisher’s exact test; s: seconds.

**Table 7 ijerph-18-13016-t007:** Steps applied (self-reported) in hand hygiene.

		*n* (%)	Age*p*-Value(Effect Size)	Gender*p*-Value(Effect Size)	Field of Study*p*-Value(Effect Size)
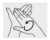	Step 1	673 (94.0)	--	--	--
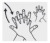	Step 2	497 (69.4)	0.040(0.09-negl)	--	0.000(0.23-moderate)
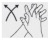	Step 3	576 (80.4)	--	--	0.002(0.16-weak)
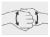	Step 4	345 (48.2)	--	0.007(0.11-weak)	0.000(0.32-moderate)
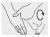	Step 5	394 (55.0)	--	--	0.000(0.30-moderate)
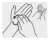	Step 6	418 (58.4)	0.008(0.11-weak)	--	0.000(0.23-moderate)

negl: negligible. “--“: not statistically significant.

**Table 8 ijerph-18-13016-t008:** Age group, gender and field of study acting as FACILITATORS for proper hand hygiene.

	Age *n* (%)	Gender *n* (%)	Field of Study *n* (%)
I have learned about HW in workshops	22–25 132 (37.3%)	Women 298 (84.2%)	Health Sciences 212 (89.9%)
Between 90–95% of microorganisms are eliminated with proper HW	22–25 88 (37.1%)	Women 195 (82.3%)	Health Sciences 156 (67.8%)
The use of a water-alcohol solution is the most effective HW method	22–25 20 (45.5%)	Women 33 (75.0%)	Health Sciences 27 (61.4%)
I wash my hands as often as necessary	22–25 128 (42.7%)	Women 232 (77.3%)	Health Sciences 187 (64.0%)
I spend between 40–60 s for the whole process of HW with water + soap	22–25 110 (42.1%)	Women 211 (80.8%)	Health Sciences 178 (70.4%)
I spend between 20–30 s for the whole process of HW with a water-alcohol solution	22–25 60 (36.6%)	Women 138 (84.1%)	Health Sciences 110 (68.8%)

HW: hand washing. s: seconds.

**Table 9 ijerph-18-13016-t009:** Age group, gender and field of study acting as BARRIERS for a proper hand hygiene.

	Age *n (%)*	Gender *n (%)*	Field of Study *n (%)*
I have learned about HW in workshops	18–21 101 (28.5%)	Men 53 (15.0%)	Engineering & Architecture 3 (0.9%)
Between 90–95% of microorganisms are eliminated with proper HW	18–21 73 (69.2%)	Men 25 (21.6%)	Engineering & Architecture 11 (4.8%)
The use of a water-alcohol solution is the most effective HW method	18–21 9 (20.5%)	Men 11 (25.0%)	Arts & Humanities 2 (4.5%)
I wash my hands as often as necessary	26–29 78 (26.0%)	Men 66 (22.0%)	Arts & Humanities 16 (5.5%)
I spend between 40–60 s for the whole process of HW with water + soap	26–29 75 (28.7%)	Men 46 (17.6%)	Arts & Humanities 12 (4.7%)
I spend between 20–30 s for the whole process of HW with a water-alcohol solution	26–29 49 (29.9%)	Men 23 (14.0%)	Arts & Humanities 6 (3.8%)

HW: hand washing. s: seconds.

## Data Availability

Please contact the corresponding author (anna.espart@udl.cat) to request the data.

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
