# Peer review of "Influence of Gender, Age and Field of Study on Hand Hygiene in Young Adults: A Cross-Sectional Study in the COVID-19 Pandemic Context"

_ijerph, 2021, doi:10.3390/ijerph182413016_

Round 1
Reviewer 1 Report
The work is based on a solid experimental design and on a well-conducted statistical processing. The text describes the work completely. The study also has the merit of investigating exclusively on hand hygiene in young adults for the first time.
However, although the characteristics of the sample are very particular, not enough mention is made of them in the text. In fact, the composition of the sample is very conditioned by the way in which the questionnaire was disseminated. As the authors note, "participants shared the questionnaire with their closest acquaintances with similar sociodemographic characteristics". In particular, the diffusion technique employed maximized the number of participants, but produced a hyper-focusing effect on people with a health sciences education. The prevalence of respondents with a health sciences education (59,4%) cannot belong to a random sample. This aspect of the research should be clearly highlighted from the beginning of the paper, describing the distribution of the sample in relation to education, possibly already in the title, but at least in the "Abstract" and in the "Implications".
My judgment is to accept the paper for publication after the required corrections have been made.
15-16: The identification of the main barriers and facilitators of HH in young adults (aged 18-29 years old), will contribute to the better planning of HH training and its posterior success.
The comma between subject and complement is not appropriate, I recommend eliminating it
17: Since they characterize the sample, add the sociodemographic characteristics of the participants.
74: … Independence …
The first letter in capital form is neither necessary nor appropriate. I would correct.
101-103: For this study, an observational, descriptive and cross-sectional design was utilized through the use of an online questionnaire, to identify the knowledge, practices, and attitudes of the study participants, related to HH.
The composition of the sample should have been conditioned by the first groups of people involved, for example colleagues of the authors. It would be useful for the characterization of the research field to indicate the initial conditions in which the dissemination of the questionnaire took place.
Tables 2, 3, 4, 5, 6, 7: In order to be clearer, please describe describe in a note the meaning of the absence of values indicated with "--".
224-226: As for the attitudes of the participants with respect to HH, it was observed that the number of times they indicated to perform it was weakly associated (p = 0.001; effect size 225 = 0.17), to the age group they belonged to.
It would be clearer and more formally appropriate: As for the attitudes of the participants with respect to HH, it was observed that the number of times they indicated to perform it was weakly associated to the age group they belonged to (p = 0.001; effect size 225 = 0.17).
371-372: Age was found to be a key factor; ….
The correlations between increasing age and knowledge, practices and attitudes are closely linked to specific age ranges, therefore, for clarity, they should be remembered.
Author Response
Dear reviewer,
thank you for your time reviewing the manuscript and for your kindly comments that help us to improve our work.
The suggested changes are included (in red letters) and justified below. Please, note that changes in blue letters refer to changes suggested by the second reviewer.
Main comments: it was included the following paragraph in the Abstract section, to highlight the over-representation of participants from the Health Science field:
Given the overrepresentation of participants from the healthcare field, it would be desirable to conduct more studies to ensure a better representation of the different educational level and field of study of the participants, in order to identify, in a more reliable way, the variables that influence HH.
In the same way, it was included the next paragraph in the “Implications” section:
Similarly, given the overrepresentation of participants from the health field, it should be taken into account that the design of strategies to improve HH will have to be specific to this sector, which already receives specific training during their education. In the case of participants trained in fields other than health, a deeper knowledge of knowledge, practices and attitudes will be required, so that the strategies designed can be adapted to their field of training.
Although the title could be modified to include the major field of study of the participants, we have preferred to leave it this way so as not to limit the interest of potential readers. Although there is overrepresentation, we believe that the study also provides data on other fields of study of the participants that may explain the tendency for participants from the Arts and Humanities or Engineering and Architecture fields to be less compliant with proper hand hygiene.
Nevertheless, we thank you for your kind recommendation.
Lines 15-16: we have deleted the comma.
Line 17 --> line 18: we included the words which analyzed, among others, the age range, gender and field of study they belonged to, in order to clarify the main sociodemographic characteristics analyzed
Line 74 --> line 82: the capital letter of “Indepence” was replaced with a lowercase letter
Lines 101-103 --> 108-113: the following paragraph was included to clarify how the participants were contacted:
Based on snowball sampling, the main author contacted people close to her, using Whatsapp® and Telegram ® contacts to participate in the study, and the participants themselves were asked to disseminated the questionnaire among their acquaintances. Likewise, content creators from different fields (related and unrelated to the health field) were contacted to publicize the study and recruit more participants.
Tables 2, 3, 4, 5, 6, 7: the words “- -“: no statistically significant were added in order to improve the table information.
Lines 224-226: (we believe these are lines 244-246 instead of 224-226) --> 256-158. The suggested modification was included. We thank you your comment.
Lines 371-372 --> 398-405: the original paragraph was replaced by:
The main implications of the study are to consider differences in age, gender, and field of study. Given that age turned out to be a key factor, and a lower level in the indicators of knowledge, practices and attitudes is identified in older age ranges (i.e., 26-29 years), this factor should be taken into account when designing strategies to improve HH indicators. Establishing Health literacy actions in the area of HH is necessary, so that they can be applied continuously and not only as one-time training, to guarantee more successful results, and therefore, to obtain better results in the reduction of cross-contamination.
In this way, we believe that the importance of the age factor in hand hygiene indicators is explicitly specified, as suggested.
Thank you.
Reviewer 2 Report
This was an interesting study but was clearly challenged by the dissemination methods of the questionnaire. The snowball sampling to largely females who had studied health leaves one with the conclusion that women who have been educated in health studies are better at washing their hands than men, and people who haven't. This is hardly groundbreaking and offers little option for interventions or solutions, as changing one's gender or educational background is unrealistic.
There was some key literature missing, in particular on targeted hygiene - cf the work of Sally Bloomfield in particular (Maillard, J.Y., Bloomfield, S.F., Courvalin, P., Essack, S.Y., Gandra, S., Gerba, C.P., Rubino, J.R. and Scott, E.A., 2020. Reducing antibiotic prescribing and addressing the global problem of antibiotic resistance by targeted hygiene in the home and everyday life settings: A position paper. American journal of infection control, 48(9), pp.1090-1099) - less important than how often you wash your hands is whether or not you wash them at the appropriate time.
It was also unclear how 'proper' hand washing was described to or by the study participants - while the anonymity is noted, the opportunity for bias in reported behaviour in line with what is expected is still high. It would have been more appropriate to put in a false step to the questionnaire to see if respondents reported what they thought was expected, rather than what they actually do. Without observation of actual behaviour or the insertion of a false step as a 'check', the self-reported results must be questioned as a limitation of the study.
There was also no reference to the extensive literature around handwashing, and the introduction of alcohol-based gel as a hand washing mechnanism during the 2009 H1N1 influenza pandemic - the point at which it became normalised in many cultures.
It was also not clear whether young adults who had not undertaken non-compulsory education were included in the study or not, and how the subjects respondents had studies related to their careers. Were these all highly-educated white collar office workers? Their need to think about hand hygiene would be very different to manual workers who literally 'get their hands dirty'. Some discussion of this in the paper would have been welcomed.
More description is needed about the channels through with the questionnaire was disseminated - particularly over more private messaging apps such as WhatsApp and Telegram. As the sample was clearly badly affected by sampling bias, this should be explained further and more awareness given earlier in the paper to the limitation this places on the study as a more generalisable investigation into the handwashing habits of all young adults.
The figure of 29-30% who faced barriers to observing good hand hygiene due to lack of equipment is very high. This result requires more attention - what were the reasons for this and how might they be overcome?
Overall, the study contained some major methodological errors which lead to rather weak and to some extent obvious results. While the difficulties of conducting research during COVID19 is appreciated, some greeted awareness of this in the earlier sections of the paper is warranted.
Author Response
Dear reviewer,
thank you for your time reviewing the manuscript and for your comments that help us to improve our work.
The suggested modifications are included (in blue letters) and justified below. Please, note that changes in red letters refer to modifications also suggested by the other reviewer.
Comment #1 (about the snowball sampling and the interventions related to gender and field of study).
To clarify that we worked following a "snowball" strategy when contacting participants, we have included the following sentence in the "Materials and Methods" section (also required by the other reviewer):
Based on snowball sampling, the main author contacted people close to her, using Whatsapp® and Telegram ® contacts to participate in the study, and the participants themselves were asked to disseminated the questionnaire among their acquaintances. Likewise, content creators from different fields (related and unrelated to the health field) were contacted to publicize the study and recruit more participants.
Related to the little option for interventions about the gender or the field of study acting as barriers for a proper hand hygiene, we included the following paragraph in the “Conclusions” section:
Given that there are important differences between the both genders, as well as depending on the field of study, it would be necessary to explore the best strategies to improve indicators related to correct HH (i.e., knowledge, practices and attitudes) in men, and in those not trained in the field of Health Sciences.
Comment #2 (about the missing literature).
We thank you the suggested reference. We included it and also included the following sentence in the “Introduction” section:
But a correct HH not only help us in the prevention of SARS-Co-V-2 disease, but it is also essential to avoid other diseases that may required antibiotic treatment; so that the usual practice of hand hygiene also contributes to a lower number of cases of antibiotic resistance [10].
Comment #3 (about the including a false step to the questionnaire).
It is true that including a false step to the questionnaire could provide interesting information about if the participants responded consciously to the correct answer or not. However, our intention was to determine whether the participants identify their knowledge as adequate and to compare their answers with those obtained in the last five questions of the "Self-knowledge" section. Thus, although a high percentage indicated that they did perform a correct HH, the percentages of correct answers in the last five questions allowed us to identify that their perception of their knowledge and the knowledge demonstrated did not coincide.
We appreciate your comment and we have included the following sentence in the "Limitations" section:
In future studies, it would be interesting, for example, to include a question about HH knowledge that includes a false answer to determine whether the participants select it as correct consciously or by random, and thus better discriminate the possible knowledge they possess.
Comment #4 (about the 2009 H1N1).
The following sentence (with a new reference, numbered as 10) was included in the “Introduction” section:
, which was normalized as an effective method to a correct HH, since the 2009 H1N1 influenza pandemic [11].
Comment #5 (about the young adults who had not undertaken non-compulsory education).
The following paragraph has been included in the “Discussion” section, in order to clarify the possibility that some participants do not identify themselves with any of the areas of study:
There are 7% of participants who have not indicated the field of study; this is very possibly due to the fact that there is no specific training area in compulsory education. Despite this, they would only represent 50 participants of the 200 who have indicated that their highest level of education is compulsory education. That is why the rest of the participants (150) would identify themselves within one of the study areas, either because they have completed some specific training at some point or because their work area that they identify is included within one of these proposed areas.
Comment #6 (about the channels used to disseminate the questionnaire).
We included in the “Materials and Methods” section the following paragraph, explaining the method used to disseminate the questionnaire through Whatsapp, Telegram and the profiles of some content creators:
Based on snowball sampling, the main author contacted people close to her, using Whatsapp® and Telegram ® contacts to participate in the study, and the participants themselves were asked to disseminated the questionnaire among their acquaintances. Likewise, content creators from different fields (related and unrelated to the health field) were contacted to publicize the study and recruit more participants.
Likewise, as we explained in this section, it was applied a convenience sampling method, which do not allow obtain a representative sample, but allow us to recover information about knowledge, habits and opinions. The fact that we registered the answers of more than 700 respondents, permit us to know the main tendencies about the variables analyzed in the study.
Comment #7 (about the lack of equipment to a proper HH).
We included the following paragraph in the “Discussion” section:
On the contrary, a significant percentage indicated that the two main reasons why they do not carry out HH is in this order "I do not remember" and because "I do not have the necessary material." In the context of the COVID-19 pandemic, in which the presence of alcoholic solution is common in any place, it is possible that the participants who have indicated that they lack the necessary material for a good HH are those who consider that water and soap are the necessary materials for the correct HH and therefore, not being able to carry out hygiene with soap and water in all places, they consider that they lack the necessary material.
We hope that with these modifications, our work improves and becomes clearer to readers.
Thank you for your suggestions.